# Detection and Characterization of Antibacterial Siderophores Secreted by Endophytic Fungi from *Cymbidium aloifolium*

**DOI:** 10.3390/biom10101412

**Published:** 2020-10-06

**Authors:** Srinivas Chowdappa, Shubha Jagannath, Narasimhamurthy Konappa, Arakere C. Udayashankar, Sudisha Jogaiah

**Affiliations:** 1Fungal Metabolites Research Laboratory, Department of Microbiology and Biotechnology, Jnana Bharathi Campus, Bangalore University, Bangalore, Karnataka 560 056, India; shubha.jagannath@gmail.com; 2Department of Studies in Biotechnology, University of Mysore, Manasagangotri, Mysuru, Karnataka 570 006, India; n.murthy10@gmail.com (N.K.); ac.uday@gmail.com (A.C.U.); 3Laboratory of Plant Healthcare and Diagnostics, PG Department of Biotechnology and Microbiology, Karnataka University, Dharwad, Karnataka 570 006, India

**Keywords:** siderophore, endophytic fungi, orchid, CAS agar, *Penicillium chrysogenum*, plant pathogens, bioformulation

## Abstract

Endophytic fungi from orchid plants are reported to secrete secondary metabolites which include bioactive antimicrobial siderophores. In this study endophytic fungi capable of secreting siderophores were isolated from *Cymbidium aloifolium*, a medicinal orchid plant. The isolated extracellular siderophores from orchidaceous fungi act as chelating agents forming soluble complexes with Fe^3+^. The 60% endophytic fungi of *Cymbidium aloifolium* produced hydroxamate siderophore on CAS agar. The highest siderophore percentage was 57% in *Penicillium chrysogenum* (CAL1), 49% in *Aspergillus sydowii* (CAR12), 46% in *Aspergillus terreus* (CAR14) by CAS liquid assay. The optimum culture parameters for siderophore production were 30 °C, pH 6.5, maltose and ammonium nitrate and the highest resulting siderophore content was 73% in *P. chrysogenum*. The total protein content of solvent-purified siderophore increased four-fold compared with crude filtrate. The percent Fe^3+^ scavenged was detected by atomic absorption spectra analysis and the highest scavenging value was 83% by *P. chrysogenum*. Thin layer chromatography of purified *P. chrysogenum* siderophore showed a wine-colored spot with R_f_ value of 0.54. HPLC peaks with R_t_s of 10.5 and 12.5 min were obtained for iron-free and iron-bound *P. chrysogenum* siderophore, respectively. The iron-free *P. chrysogenum* siderophore revealed an exact mass-to-charge ratio (*m*/*z*) of 400.46 and iron-bound *P. chrysogenum* siderophore revealed a *m*/*z* of 453.35. The solvent-extracted siderophores inhibited the virulent plant pathogens *Ralstonia solanacearum*, that causes bacterial wilt in groundnut and *Xanthomonas oryzae* pv*. oryzae* which causes bacterial blight disease in rice. Thus, bioactive siderophore-producing endophytic *P. chrysogenum* can be exploited in the form of formulations for development of resistance against other phytopathogens in crop plants.

## 1. Introduction

The plants and microbes in soil require micronutrient metals such as nickel, copper, zinc and iron but the bio-availability of these elements is often inadequate under environmental conditions [1]. The adverse environmental conditions lead to decreased bio-accessibility of Fe (III) due to synthesis of non-soluble oxyhydroxide phases [2]. Iron deficit leads to chlorosis and reduced metabolic activity and biomass in plants. Plants and microbial species have developed a chelation approach to stimulate metal availability during stress situations [1,3]. The fungal species in soil and plant endophytes synthesize hydroxamate and carboxylate type siderophores of several classes—coprogens, fusarinines and ferrichromes [4,5]. The siderophores produced by fungi play a significant role in transporting iron to plants, bacterial and actinomycetes members [6,7,8]. To date ~500 diverse siderophores have been documented [9]. A majority of the siderophores are reported to be endophytic in nature, being found in various microbial sources including mycorrhiza and orchidaceous fungi [8,10,11]. However, the nature of siderophores is different among microbes, for instance the bacterial hydroxamates are made up of hydroxylated and acylated alkylamines, while fungal hydroxamates are composed of hydroxylated and acylated ornithine groups. Most of the hydroxamate groups of siderophores of fungal origin consist of C (=O) N-(OH)-R, where R is either an amino acid or a derivative. The two oxygen molecules from each hydroxamate group act as chelating agents forming hexadentate octahedral complexes with Fe^3+^ that exhibit high stability constants. They also act as competitors for Fe thereby reducing the amount of Fe available to plant pathogens. This Fe deficiency inhibits the growth of phytopathogens by inhibiting the synthesis of nucleic acids and sporulation of the pathogens.

Siderophores are reported to have many favorable applications for mankind and Mother Nature. They can be used for selective antibiotic delivery—a Trojan horse strategy by formation of sideromycins, useful in treating antibiotic-resistant bacteria—or used to treat acute iron intoxications such as haemochromatosis. They are also useful in the treatment of malaria and in the removal of transuranic elements such as aluminium and vanadium from human body [12,13], as deodorant in cosmetics, in cancer therapy [14], as bio-control agents against fish pathogens [15], documented bio-control agents against bacterial and fungal phytopathogens [7,8,16], in pulp treatment [17], bioremediation of mercury [18], to solubilize a varied array of heavy metals such as Cd, Zn, Ni, Cu, Pb and actinides like Pu, Th and U produced in industries, nuclear power stations and mining [19,20]. Siderophores are also reported to diminish oxidative stress in microorganisms [21]. The siderophores have prospects as selective investigative tools and in the delivery of antifungal drugs [22]. Siderophores are also used for the classification of microorganisms based on the type of siderophore they produce, which is known as siderophore typing or sidero-typing [23]. In recent years, research has focused on the identification and characterization of bioactive siderophores from endophytic fungi isolated from a medicinal orchid plant, *Cymbidium aloifolium*. In most parts of southern India, this orchid plant is traditionally used for the treatment of cut wounds or burns, low eye vision, as an anti-inflammatory and anti-bacteria agent, and for paralysis, chronic illness and also for fevers [24], hence, the present investigation is focused on: (i) isolation of endophytic fungi associated with different parts of *C. aloifolium*, (ii) optimization of culture conditions for the production of siderophores from endohytic fungi, (iii) detection and quantification of siderophore production from isolated endophytic fungi, (iv) identification and purification of type of siderophores and (v) the bioactivity of the siderophores against phytopathogens such as *Ralstonia solanacearum* and *Xanthomonas oryzae pv. Oryzae.*


## 2. Materials and Methods 

### 2.1. Endophytic Fungi from Cymbidium Aloifolium

Fifteen healthy orchid plants (free from lesions) were collected from different regions covering a distance of 270 km in the Western Ghats region of Karnataka (N 12°45′14.2351″, E 75°38′38.7031″) at locations such as Kemmangundi, Sringeri, Shivamogga and Chikmagalur during flowering time between March to June 2017. The plant specimens were identified as *Cymbidium aloifolium* by the Regional Ayurveda Research Institute for Metabolic Disorders, Central Council for Research in Ayurvedic Sciences, Ministry of AYUSH, Government of India. The plant specimens were labeled with the reference number RRCBI-7148. Leaves, roots and flowers were harvested from the original plant and surface sterilized using sodium hypochlorite (0.2%), following by three washes with sterile distilled water. The dried plant parts were used for isolation of endophytic fungi on Potato Dextrose Agar (PDA) plates amended with tetracycline (50 µg/mL) and sterilized plant parts incubated on PDA without tetracycline were used as appropriate controls as described by Zhu et al. [25] and the fungi identified using standard manuals [26,27]. The endophytic fungi were identified by morphological characteristics as well as using ITS region sequencing (ITS1 and ITS4 primers). The obtained sequences were assembled and blast search for their similarity or homology by NCBI. The authenticated names of the fungal species were submitted to the NCBI GenBank database. Two representative plants (reference number RRCBI-7148) were also maintained in the orchidarium at the Department of Botany, Bangalore University, Bangalore. A specimen is also maintained at the Regional Ayurveda Research Institute for Metabolic Disorders, AYUSH, Bangalore. 

### 2.2. Screening for Production of Siderophore by Endophytic Fungi

The siderophore production was assessed by placing 4 mm of 1-week old mycelia plugs on Chrome Azurol S (CAS) solid media prepared by following the modified protocol (added 22 g of PIPES/L instead of 32.24 g) of Schwyn and Neilands [28]. The fungi-inoculated plates were incubated at 30 °C under dark conditions for 5–7 days and then monitored for the appearance of an orange halo zone around the fungal disc on blue colored agar media, and the zone diameter was then measured.

### 2.3. Determination of Siderophore Concentration by CAS Liquid Assay

Iron deferrated Grimm-Allen liquid media [29] was used for growing fungi (10^6^ spores/mL, 30 °C for 15 d) and culture filtrate was used to estimate siderophore content. The filtrate (1.5 mL) added with 1.5 mL of CAS liquid solution, 10 µL of shuttle solution (0.2 M 5-sulfosalicylic acid, store in dark) was incubated at room temperature for 10 min and absorbance measured at 630 nm. The siderophore content was calculated using the following formula:% Siderophore units = [(Ar − As)/Ar]
where A_r_ = Reference absorbance at 630 nm, As = Absorbance of sample at 630 nm. Blank: culture medium; Reference: culture medium with CAS liquid solution and shuttle solution.

### 2.4. Detection of Siderophore Type

The fungal culture filtrates positive for siderophore production were subjected to a FeCl_3_ test. Two mL of FeCl_3_ solution (2%) was added to 1 mL of fungal culture filtrate and scanned from 300 nm to 600 nm using an UV-Vis spectrophotometer (UV-160A, Shimadzu, Kyoto, Japan). A peak between 420–450 nm in ferrated siderophores indicates a hydroxamate type of siderophore and a peak at 495 nm indicates a catecholate type of siderophore [30]. For the tetrazolium test two drops of 2 N NaOH was added to a pinch of iodonitrotetrazolium (INT) and 1 mL of culture filtrate was added and observed for the appearance of a deep colour to determine the presence of hydroxamate siderophores [31]. Briefly, the Csaky assay consists of 1 mL of culture filtrate hydrolyzed with 1 mL of 6 N H_2_SO_4_ in a boiling water bath for 6 h or at 130 °C for 30 min. The solution was then buffered by adding 3 mL of 35% sodium acetate solution. Further, the resultant solution was supplemented with 1 mL of sulfanilic acid solution followed by 0.5 mL of iodine solution. Finally, 1 mL of α-naphthylamine solution was added and the total volume was adjusted to 10 mL with sterile distilled water. The reaction solution was incubated for 20–30 min at room temperature and monitored for changes in the reaction solution color. In parallel culture media was used as blank [32] to detect the type of siderophore.

### 2.5. Optimization of Culture Parameters for Maximum Siderophore Production

The endophytic fungi exhibiting higher siderophore content were subjected to optimization studies. An inoculum of 10^6^ spores/mL was inoculated into 15 mL of iron deferrated media. The effect of growth media, incubation period, temperature, pH, carbon source, nitrogen source was determined. The percent siderophore content produced in each parameter was determined using a CAS liquid assay.

### 2.6. Production and Extraction of Siderophore Produced by Endophytic Fungi

The endophytic fungi producing higher amounts of siderophore were grown in 500 mL optimized deferrated media, incubated in the dark at 120 rpm. The siderophore-rich culture filtrate was extracted following the modified procedure of Jalal and Helm [33].

### 2.7. Estimation of Total Protein Concentration in Crude and Solvent Purified Siderophore Extract

The total protein concentration in crude and solvent purified siderophore extract was estimated by the Lowry method [34].

### 2.8. Agar Well Diffusion of Crude and Solvent Extracted Siderophore on CAS Agar

The crude and partially purified siderophore extract (25 µL) was detected by agar well diffusion on CAS agar media, by detecting the development of a orange halo around the well.

### 2.9. Atomic Absorption Spectra (AAS) Analysis of Solvent Extracted Siderophore

Partially purified endophytic fungi siderophore samples suspended in 2% FeCl_3_ solution were used as samples. FeCl_3_ (2%) in sterile distilled water was taken as control. The amount of dissolved Fe^3+^ in each sample was evaluated using an AA240FS fast sequential atomic absorption spectrometer (Agilent Technologies, California, USA) equipped with an iron Lumina halo cathode lamp [35]. The percent Fe^3+^ present in siderophore suspended sample was determined using the formula:Total Fe^3+^% with siderophore treatment = % Fe^3+^ in Siderophore treated sample by AAS × 100% Fe in Control by AAS

The percentage of Fe^3+^ scavenged by fungal siderophores was determined by the formula:% Fe^3+^ scavenged = 100 − % Fe^3+^ in siderophore sample with reference to control.

### 2.10. Purification and Characterization of Siderophore

An extracted siderophore sample (CAL1) exhibiting higher siderophore content with good biological activity was purified using Amberlite XAD-400 chromatography [36]; purified siderophore was then spotted on silica gel thin layer chromatography (TLC) plates and eluted with chloroform and methyl alcohol (90:10 *v*/*v*). The plates were dried, sprayed with 0.1 M FeCl_3_ in 0.1 N HCl, allowed to dry [37] and observed for the formation of spots.

### 2.11. Liquid Chromatography Election Spray Ionization-Mass Spectrometry

The profiling of the LC-ESI mass spectrum of ferrated and deferrated siderophore sample (CAL1) was done at the Central Research Facilities, Indian Institute of Science, Bangalore [38]. The MS system used for profiling was a maxis impact HD ESI QTOF high resolution mass spectrometer (Bruker Daltonics, Bremen, Germany). The siderophore sample was diluted with acetonitrile and water (1:1 *v*/*v*) with 0.5% acetic acid and filtered. The mobile phase used was acetonitrile and water (1:1 *v*/*v*) containing 0.5% acetic acid. Mass was measured in the range from *m*/*z* 50–1200 with a flow rate of 0.3 mL/min. The parameters used were electron spray ionization (ESI), ion trap analyzer, ion polarity positive set nebulizer 1.8 bars, focus active capillary 3500v and dry heater 180 °C.

### 2.12. Antibacterial Activity of Siderophores Against Plant Pathogenic Bacteria

The antibacterial activity of extracted fungal siderophores was determined by a disc diffusion assay [39]. Ampicillin (10 µg/mL) was used as positive control and 10% sterile media as negative control. The fungal siderophores was tested against virulent plant pathogens such as *Ralstonia solanacearum* and *Xanthomonas oryzae* pv*. oryzae*. The pure cultures were obtained from the Department of Microbiology and Biotechnology, Bangalore University, Jnana Bharathi Campus, Bangalore.

### 2.13. Statistical Analysis

The tests were done in triplicates and results were stated as mean ± Standard deviation. The analysis was performed with ANOVA and Duncan Multiple Range Test (DMRT) test was carried out using SPSS software (version 20, Tokyo, Japan). A value of *p ≤* 0.05 was considered a significant difference. 

## 3. Results

### 3.1. Screening for Production of Siderophore by Endophytic Fungi

The endophytic fungi obtained from different parts of *C. aloifolium* like flowers, leaves and roots were screened for siderophore production on CAS agar. Among endophytes, 60% of fungi produced siderophores (Figure 1). Among endophytic fungi, the highest diameter of the halo zone was 14.33 ± 2.08 mm produced by *P. chrysogenum* (CAL1) followed by other fungal isolates (Table 1).

### 3.2. Determination of Siderophore Concentration by CAS Liquid Assay

The percent siderophore content among endophytic fungi ranged from a high of 56.6% by *P. chrysogenum* (CAL1) to a low of 10.35% by *C. truncatum* (CAL5). The endophytic fungi *A. sydowii* (CAR12) produced 48.94% and *A. terreus* (CAR14) produced 46.39%, respectively (Table 2, Appendix A). The endophytic fungi producing higher siderophore content were identified at a molecular level by subjecting them to partial ITS sequencing using ITS1 and ITS4 primers. The obtained partial ITS sequences were submitted to GenBank with NCBI accession numbers KX553900 (*Penicillium chrysogenum*, CAL1), KX553901 (*Aspergillus sydowii,* CAR12) and KX553902 (*Aspergillus terreus,* CAR14), respectively. Further, the phylogentic dendrogram tree was plotted to determine the taxonomic status of the isolated endophytic fungi in relation to existing fungi of NCBI database (Appendix A).

### 3.3. FeCl_3,_ Tetrazolium and Csaky Test

The endophytic fungal culture filtrate with 2% FeCl_3_ exhibited a broad peak range between 420–450 nm indicating hydroxamate type siderophores (Appendix A). The tetrazolium test recorded the instant appearance of a deep colour indicating hydroxamate type siderophores (Appendix A). The Csaky assay recorded the presence of hydroxamate type of siderophore by the development of colour (Appendix A). This method is sensitive and specific for hydroxamate type siderophores.

### 3.4. Optimization of Culture parameters for Maximum Siderophore Production by Endophytic Fungi

The endophytic fungal cultures *C. gloeosporioides* (CAR4), *Xylaria* sp. (CAR6), *A. sydowii* (CAR12), *A. terreus* (CAR14) and *P. chrysogenum* (CAL1) were selected for optimization studies. The siderophore production was found to be higher in Grimm Allen (GA) media in comparison to MM9 media among all fungal endophytic cultures studied (Figure 2a). 

A gradual increase in the percent siderophore content was observed on the 5th day, 10th day and 15th day in all fungal cultures but the content decreased slightly on the 20th day. The highest production recorded was on the 15th day for all endophytic cultures (Figure 2b). The siderophore content varied with incubation temperature, being optimum at 30 °C, lesser at 35 °C and least at 25 °C and no siderophore production was observed at 20 °C or 40 °C (Figure 2c). The pH of the growth medium plays a key role in the solubility of iron and thus its accessibility to endophytic fungi. The maximum siderophore content was detected at pH 6.5, slightly decreased at pH 6 and pH 7; no siderophore was detected at pH 5.5 and pH 7.5, respectively (Figure 2d). The siderophore production was detected with three sugars—maltose, fructose and glucose. Among these, maltose exhibited a stimulatory effect on siderophore production followed by glucose and fructose (Figure 2e). Different nitrogen sources influenced siderophore production and highest was detected with ammonium nitrate as the nitrogen source, followed by sodium nitrate. The use of urea as nitrogen source resulted in decreased siderophore production (Figure 2f).

### 3.5. Production and Extraction of Siderophore

The siderophore quantity with optimized parameters was estimated by CAS liquid assay. The highest siderophore content was detected in *P. chrysogenum* (CAL1)—72.69% followed by *A. sydowii* (CAR12)—65.29%, *A. terreus* (CAR14)—59.04%, *C. gloeosporioides* (CAR4)—46.52% and *Xylaria* sp. (CAR6)—29.46% (Appendix A). The culture filtrates of CAL1, CAR12 and CAR14 producing the highest siderophore content were subjected to extraction with chloroform:phenol: ether:water and purified.

### 3.6. Estimation of Total Protein Content

The total protein content of solvent-purified siderophore sample was found to be 3.52 mg/mL for *P. chrysogenum* (CAL1) followed by 3.51 mg/mL for *A. sydowii* (CAR12) and 3.27 mg/mL for *A. terreus* (CAR14), respectively, whereas, the total protein content of crude siderophore filtrate was much less compared to purified samples viz., 0.79 mg/mL by CAL1, 0.735 mg/mL by CAR12 and 0.29 mg/mL by CAR 14 (Figure 3). The four-fold increased protein content in solvent extracted siderophore sample indicated the effectual purification of siderophore.

### 3.7. Agar Well Diffusion of Crude and Solvent Extracted Siderophore on CAS Agar

The purified siderophore extract from *P. chrysogenum* (CAL1), *A. sydowii* (CAR12) and *A. terreus* (CAR14) produced the largest diameter and clearest orange halo zones around the well on CAS agar media in comparison to crude extract (Appendix A). 

### 3.8. Atomic Absorption Spectra (AAS) Analysis

The percent Fe^3+^ scavenged was 83.12% for *P. chrysogenum* (CAL1) followed by 73.34% for *A. terreus* (CAR14) and 71.12% for *A. sydowii* (CAR12) respectively (Table 3).

### 3.9. Purification of Penicillium Chrysogenum (CAL1) Siderophore

The *P. chrysogenum* (CAL1) siderophore exhibited the highest Fe^3+^ scavenging property, hence it was further purified using Amberlite XAD-400 chromatography and further used for TLC and LC-ESI-MS analysis.

### 3.10. Thin Layer Chromatography of Purified Siderophore

The formation of a wine coloured spot with a R_f_ value of 0.54 indicated a hydroxamate type of siderophore (Appendix A).

### 3.11. High-Performance Liquid Chromatography (HPLC) and Liquid Chromatography Electron Spray Ionization Mass Spectrometry (LC-ESI-MS)

HPLC peaks at a R_t_ of 10.5 min was recorded for iron-free and a peak with a R_t_ at 12.5 min was recorded for iron-bound *P. chrysogenum* (CAL1) siderophore (Figure 4a,b). The iron-free siderophore *P. chrysogenum* (CAL1) siderophore revealed an exact mass-to-charge ratio (*m*/*z*) of 400.46 and iron-bound *P. chrysogenum* (CAL1) siderophore revealed a *m*/*z* of 453.35, respectively (Figure 5a,b).

### 3.12. Antibacterial Activity of Fungal Siderophores on Plant Pathogenic Bacteria Ralstonia Solanacearum and Xanthomonas oryzae pv. Oryzae

The solvent-extracted siderophores obtained from *A. sydowii* (CAR12), *P. chrysogenum* (CAL1), *A. terreus* (CAR14) inhibited different virulent *R. solanacearum* (Table 4) and *Xanthomonas oryzae* pv*. oryzae* isolates (Table 5). The siderophore from *P. chrysogenum* (CAL1) exhibited the highest zone of inhibition in comparison to *A. sydowii* (CAR12) and *A. terreus* (CAR14) respectively (Figure 6a,b). 

## 4. Discussion

Microbial siderophore synthesizers have been the subject of more interest because of the potential application of these chelators in agriculture and also in clinical applications [40,41]. The siderophores have been utilized in the initial stages of lignocellulose depolymerization of wood cell wall by fungi [42]. Hence, the present study was focused on the production, optimization, purification and characterization of siderophores produced by endophytic fungi of *C. aloifolium* and their antibacterial activity against plant pathogens. The siderophore production by endophytic fungi of *C. aloifolium* was detected on CAS agar based on the greater affinity of siderophores towards ferric iron. 

Among endophytes, 60% endophytic fungi showed an orange halo zone of 4 mm around mycelia plugs, indicating their ability to produce siderophores. The findings in this study are in agreement with Aramsirirujiwet et al. [43] who reported that 58% of the endophytic fungi isolated from healthy tissues of *Hottuynia cordata* produced siderophores. In the present study, the diameter of halo zones varied with different fungal cultures and the highest diameter was recorded by *P. chrysogenum*—CAL1 (14.33 mm), this is in agreement with Hordt et al. [44] who reported that *P. chrysogenum* isolated from soil produced different siderophores such as fusarinines, dimerum acid, mono- and dihydroxamate siderophores which enhanced iron uptake in cucumber and maize plants. The results were also supported by Baakza et al. [45] who reported the production of siderophores in endophytic fungi associated with marine and terrestrial habitats.

The percent siderophore content among fungi ranged from a highest value of 56.6% to a lowest of 10.35%. The highest siderophore percentage was observed in *P. chrysogenum* (CAL1) followed by other endophytic fungi. The results obtained were directly proportional to the halo zone produced in a CAS agar test and are in accordance with the findings of Calvente et al. [46]. On the contrary, Schwyn and Neilands [28] reported that quantification of siderophores was possible only by CAS liquid assay as it was not accurate to access siderophore quantity on CAS solid agar. 

The fungal siderophore culture filtrate with 2% FeCl_3_ exhibited a broad peak between 420–450 nm indicating a hydroxamate type of siderophore. This result agrees with Baakza et al. [45] who reported that a peak at 425–450 nm indicated a hydroxamate type of siderophores. The instant formation of a deep colour in the tetrazolium test indicated a hydroxamate type of siderophore. This is in agreement with Dave et al. [47] who reported that the tetrazolium test is based on the ability of hydroxamic acids in presence of strong alkali to reduce tetrazolium salts by hydrolysis of the hydroxamate group resulting in appearance of a deep colour. The Csaky assay detects the presence of hydroxamate type siderophores by the development of colour. This method is sensitive and specific for hydroxamate type siderophores [32]. The Csaky test for hydroxylamine is thus widely used as a standard method for the detection and assay of hydroxamic acid type siderophores; it is a modification of the Bloom iodine oxidation method. The product of this assay is nitrite [48].

The various culture parameters influenced the production of siderophores by endophytic fungi of *C. aloifolium*. The test media used for optimization studies were iron deferrated. Volker and Wolf [49] have previously reported that siderophore production takes place only under iron-limited conditions to scavenge ferric ions. The fungal growth media significantly influenced siderophore production; among Grimm Allen (GA) and MM9 media tested, the fungi grown in GA media produced a greater amount of siderophores. Similar results were documented by Grimm and Allen [29] who reported that fungi belonging to the *Ascomycota* and *Basidiomycota* produced siderophores when grown in Grimm Allen media. The incubation period and temperature considerably influenced siderophore production; these findings are in accordance with Dave and Dube [50] who have reported that maximum siderophore production by fungal cultures occurred at 30° C with an incubation period of 15 days. The pH of the growth medium has an important role in the solubility of iron and thus its availability to the fungi. The maximum siderophore content was detected at pH 6.5, and slightly decreased at pH 6 and pH 7 for all the endophytic fungi tested, but Dave and Dube [50] have reported maximum siderophore production by fungi at pH 7.

The siderophore production was detected with maltose, fructose and glucose as carbon sources; maltose demonstrated a stimulatory effect on siderophore production when compared to glucose and fructose. This result correlates with Roy et al. [51] who reported that the carbon source influences siderophore production. The previous findings of Mahmoud and Alla [52] have shown a higher siderophore production with glucose as carbon source but Sistorm and Michilis [53] reported fructose as the best utilized carbon for siderophore production. The nitrogen sources considerably influenced siderophore production. The use of ammonium nitrate as nitrogen source enhanced siderophore production, followed by sodium nitrate, whereas urea resulted in reduced siderophore production although Tailor and Joshi [54] have reported urea as the best nitrogen source for siderophore production by *Pseudomonas fluorescens*, which was not in accord with our observation in endophytic fungi.

The results obtained in our study correlate with the findings of Aziz et al. [55] who have reported that the production of fungal siderophores in vitro is significant influenced by abiotic factors such as temperature, pH, culture media, carbon and nitrogen sources.

The siderophore production was enhanced when endophytic fungi were grown in iron- deficient media under dark conditions with shaking at 120 rpm. *P. chrysogenum* (CAL1) produced a greater siderophore content, followed by *A. sydowii* (CA12) and *A. terreus* (CAR14). The endophytic fungal culture filtrates containing siderophores were further subjected to extraction and purification by chloroform:phenol:ether:water extraction. Pidacks et al. [56] have reported that chloroform: phenol solution is a better solvent for extraction of siderophores in comparison to benzyl alcohol as the transfer of siderophores to the organic phase was easier with this solvent mixture. The *P. chrysogenum* (CAL1) siderophore sample was further purified using an Amberlite XAD-400 column as it recorded the greater siderophore production. 

The best solvent system for detection of *P. chrysogenum* (CAL1) siderophore on TLC was chloroform and methanol (90:10 *v*/*v*). The spraying with 0.1 M FeCl_3_ in 0.1 N HCl developing solution resulted in the appearance of a wine coloured spot with an R_f_ value of 0.54, confirming the hydroxamate type of the siderophores. The siderophore R_f_ value obtained in this study closely matched the R_f_ value (0.52) of bisucaberin as reported by Yoshiro et al. [57]. The molecules are separated based on their polarity as the solvent moves up the gel plate; polar compounds bind strongly with silica gel and hence move slower when compared to non-polar compounds [58].

A HPLC peak at a R_t_ of 10.5 min was obtained for iron-free and a peak at a R_t_ of 12.5 min was obtained for iron-containing *P. chrysogenum* (CAL1) siderophore. The iron-free *P. chrysogenum* (CAL1) siderophore revealed an exact mass-to-charge ratio (*m*/*z*) of 400.46 and iron-bound *P. chrysogenum* (CAL1) siderophore revealed a *m*/*z* of 453.35. The mass of siderophore obtained in the present findings was compared with a list of sixty previously reported siderophores compiled in a review article by Pluhacek et al. [59]. The mass of *P. chrysogenum* (CAL1) closely matched with mass of 454.1509 reported for iron-bound bisucaberin (C_18_H_33_N_4_O_6_) which was previously reported in *Vibrio alginolyticus* B522, *Shewanella algae* B516 and *Vibrio salmonicida* [60,61]. In this study, the relative mass error of iron-bound bisucaberin was 0.8 ppm in comparison with the findings of Bottcher and Clardy [60]. The bisucaberin siderophore has also been previously reported from a marine bacterium *Alteromonas haloplanktis* [62]. The molecular constituents of bisucaberin are identical with other siderophores such as trihydroxamate ferrioxamine E, ferrioxamine G, ferrioxamine B, ferrioxamine D_1_ [63]. Bisucaberin is also structurally related to alcaligin from *Alcaligenes xylosoxidans*, *Bordetella pertussis* and *B. bronchiseptica* [64,65]. The bisucaberin siderophore obtained from *Alteromonas haloplanktis* has been previously reported to sensitize tumor cells to macrophage-mediated cytolysis thus proving a potential in cancer treatment [66].

The total protein content in solvent-purified siderophore extract was significantly (five-fold) higher compared to crude siderophore extract. The present findings are in accordance with Saxena et al. [67] who have reported that iron binding proteins are produced under iron-deficient conditions. The siderophore content in crude and purified siderophore extract was detected on CAS agar media. The purified siderophore extract of *P. chrysogenum* (CAL1), *A. sydowii* (CAR12) and *A. terreus* (CAR14) produced clear orange halo zones around the wells when compared to crude extract. This result agrees with Calvente et al. [46] who have reported that the intensity of halo zone production was proportional to the siderophore concentration. 

The percent Fe^3+^ scavenged was determined by atomic absorption spectrometry (AAS); 83.12% of Fe^3+^ was scavenged by *P. chrysogenum* (CAL1), 73.34% by *A. terreus* (CAR14) and 71.12% by *A. sydowii.* The reduction in the percent Fe^3+^ in siderophore treated 2% FeCl_3_ solution indicated a high affinity of the fungal siderophores for ferric ions. The present finding agrees with Haas [4] who reported siderophores synthesized by fungi help in the uptake of iron and also its storage. The present studies also agree with the findings of Miethke and Marahiel [14] who reported siderophore- based high-affinity iron acquisition by bacteria and fungi. These results are also in agreement with Hider and Kong [3] and Saha et al. [68] who reported siderophores produced by microbes have extraordinarily high affinity for ferric ions and tend to use negatively charged oxygens as coordinating, donor atoms. Microbial siderophore producers have received greater consideration because of significant utilization of these chelators in agriculture and also clinical applications [40,41]. Chua et al. [69] reported siderophores are used for removal of non-transferrin bound iron in serum which is an outcome of chemotherapy during cancer treatment. The findings of Miethke and Marahiel [14] have reported that siderophores can potentially be used as iron chelators in cancer treatment, e.g., in drugs like -trensox, tachpyridine, dexrazoxane and desferrithiocin desferriexochelins. 

The siderophores produced by endophytic fungi—*P. chrysogenum* (CAL1), *A. terreus* (CAR14) and *A. sydowii* (CAR12)—exhibited antibacterial activity against the potent plant pathogens *R. solanacearum*, a causative agent of bacterial wilt in groundnut and *Xanthomonas oryzae* pv*. oryzae*, a causative agent of bacterial wilt in rice. The siderophore produced by *P. chrysogenum* recorded a higher zone of inhibition against the plant pathogens tested. The present finding proves that fungal siderophores are potent agents which can be used against plant pathogens [70]. These results suggest that siderophores are produced by microorganisms during iron-limiting conditions sequester iron (III), thus making it unavailable to the pathogen [71,72,73]. The external application of siderophores utilizes iron, thereby depleting availability of iron to the pathogen, hence enabling killing of plant pathogens [14,74]. Earlier findings have reported the use of siderophores in controlling few pathogenic fungi such as *Pythium ultimum*, *Sclerotinia sclerotiorum* and *Phytophthora parasitica* causing diseases in plants [75,76].

## 5. Conclusions

Taken together, the present study focuses on the comparative analysis of siderophore production from various endophytic fungi isolated from *C. aloifolium*. The hydroxamate type siderophore isolated from *P. crysogenum* exhibited strong antibacterial properties against major virulent phytopathogens, thereby protecting groundnut and rice. Further, the bioactive siderophore producing endophytes can be exploited in the form of formulations for the development of resistance against other phytopathogens in crop plants. 

## Figures and Tables

**Figure 1 biomolecules-10-01412-f001:**
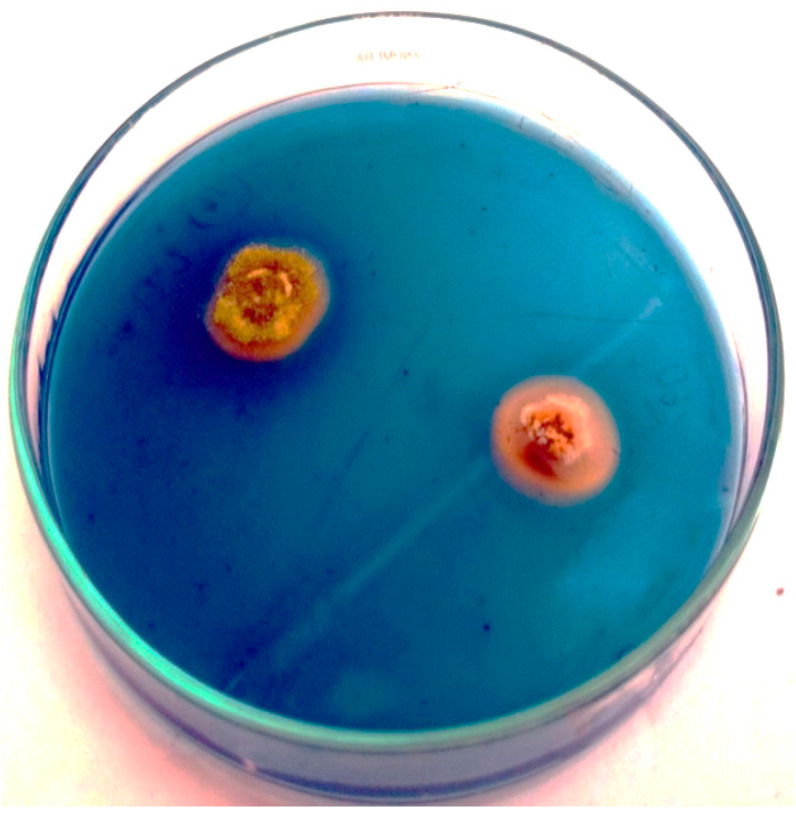
Endophytic fungi exhibiting an orange halo on Chrome Azurol S (CAS) agar, indicating siderophore production.

**Figure 2 biomolecules-10-01412-f002:**
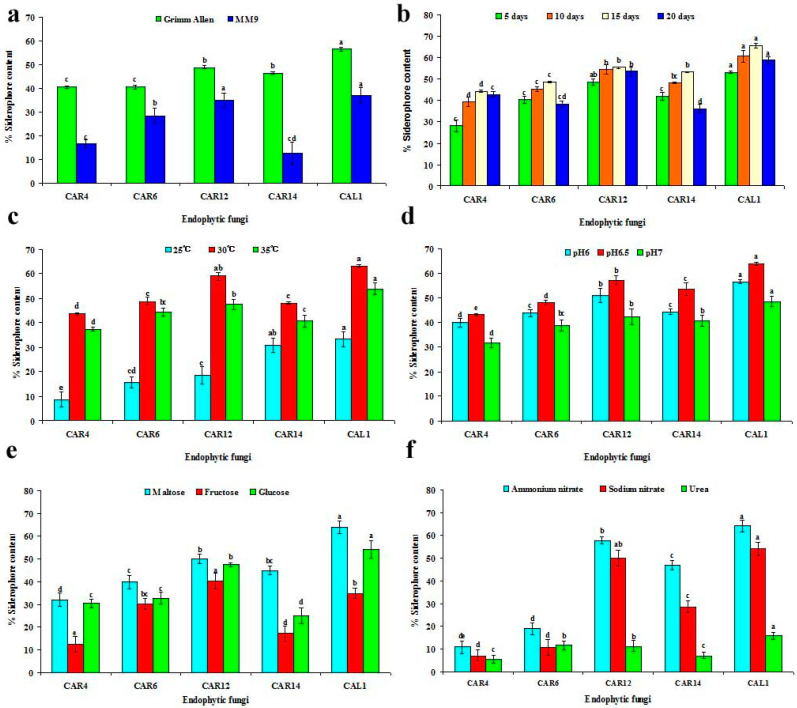
Effect of different culture parameters on siderophore production. (**a**) Effect of media, (**b**) Effect of incubation period, (**c**). Effect of temperature, (**d**) Effect of pH, (**e**) Effect of carbon sources, (**f**) Effect of nitrogen sources. Mean values followed by same letters are not significantly different according to DMRT at *p* ≤ 0.05.

**Figure 3 biomolecules-10-01412-f003:**
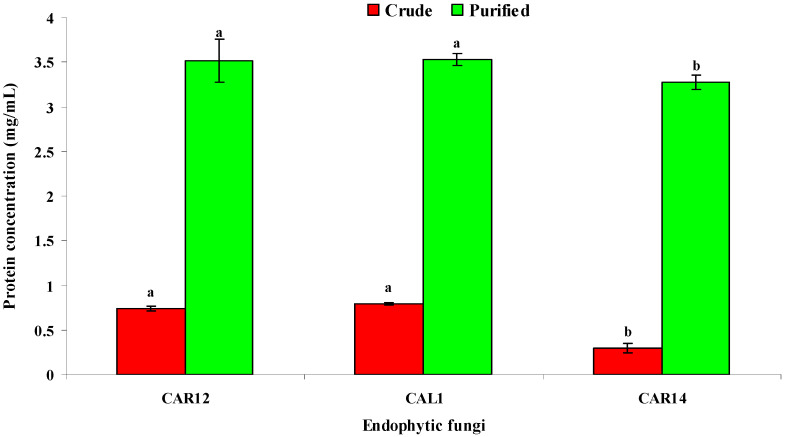
Total protein content in the crude and purified siderophore extract by *Penicillium chrysogenum* (CAL1). Treatment means annotated above by the same letter are not significantly different according to DMRT at *p* ≤ 0.05.

**Figure 4 biomolecules-10-01412-f004:**
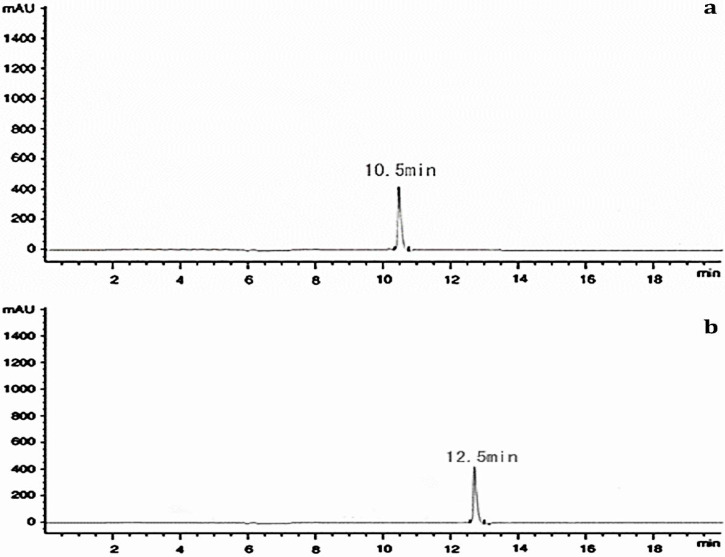
High-performance liquid chromatography (HPLC) chromatograms of *Penicillium chrysogenum* (CAL1) purified siderophore. (**a**) iron-free siderophore, (**b**) iron-containing siderophore.

**Figure 5 biomolecules-10-01412-f005:**
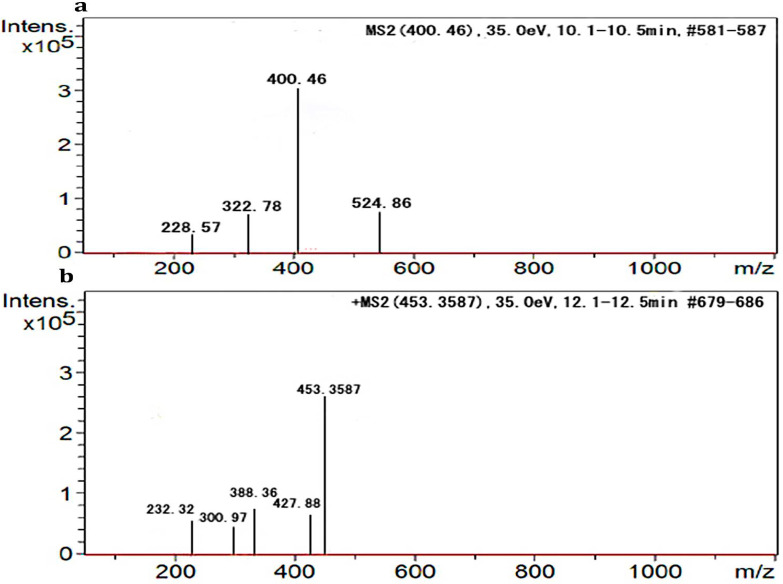
Electron spray ionization Mass spectroscopy (ESI-MS) spectrum of *Penicillium chrysogenum* (CAL1) purified siderophore (**a**) iron-free siderophore, (**b**) iron-containing siderophore.

**Figure 6 biomolecules-10-01412-f006:**
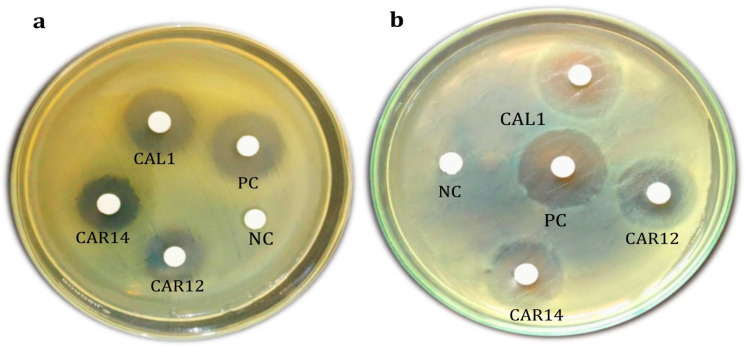
Antibacterial activity of endophytic fungal siderophores on plant pathogens (**a**) *Ralstonia solanacearum (***b**) *Xanthomonas oryzae* pv*. oryzae.*

**Table 1 biomolecules-10-01412-t001:** Screening for siderophore production by endophytic fungi of *Cymbidium aloifolium* on Chrome Azurol S Agar (CAS).

Sl. No	Isolates	Endophytic Fungi	Source	Yellow-Orange Zone on CAS	Mean of Halo Diameter in mm ± SD; *n* = 3
1	CAR 1	*Aspergillus japonicus*	**Root**	Positive	6.33 ± 0.57 ^b^
2	CAR 2	*Curvularia lunata*		Negative	0 ± 0 ^a^
3	CAR 3	*Nigrospora* sp.		Negative	0 ± 0 ^a^
4	CAR 4	*C. gloeosporioides*		Positive	9.66 ± 0.57 ^c^
5	CAR 5	*Trichoderma* sp.		Positive	7.33 ± 1.15 ^b^
6	CAR 6	*Xylaria* sp.		Positive	9.33 ± 2.08 ^c^
7	CAR 7	*Rhizoctonia* sp.		Negative	0 ± 0 ^a^
8	CAR8	*F. chlamydosporum*		Negative	0 ± 0 ^a^
9	CAR 9	*Penicillium* *citrinum*		Positive	9.33 ± 1.52 ^c^
10	CAR10	*Helminthosporium* sp.		Negative	0 ± 0 ^a^
11	CAR 11	*Curvularia* sp.		Positive	10 ± 1 ^c^
12	CAR 12	*Aspergillus sydowii*		Positive	12.33 ± 0.57 ^e^
13	CAR 13	*Cladosporium* sp.		Positive	7 ± 1 ^b^
14	CAR 14	*Aspergillus terreus*		Positive	12 ± 0 ^de^
15	CAR 15	*Alternaria* *alternata*		Positive	6.66 ± 0.57 ^b^
16	CAR 16	*Fusarium oxysporum*		Negative	0 ± 0 ^a^
17	CAL1	*P. chrysogenum*	**Leaf**	Positive	14.33 ± 2.08 ^f^
18	CAL2	*Aspergillus sydowii*		Positive	7.66 ± 1.52 ^b^
19	CAL3	*Trichoderma* sp.		Negative	0 ± 0 ^a^
20	CAL4	*Rhizoctonia* sp.		Positive	7.33 ± 1.15 ^b^
21	CAL5	*Curvularia lunata*		Negative	0 ± 0 ^a^
22	CAL6	*Penicillium citrinum*		Negative	0 ± 0 ^a^
23	CAL7	*C.* *truncatum*		Positive	7 ± 1 ^b^
24	CAL8	*Alternaria alternata*		Positive	6.66 ± 0.57 ^b^
25	CAL9	*Bipolaris* sp.		Negative	0 ± 0 ^a^
26	CAF1	*Fusarium* *oxysporum*	**Flower**	Positive	7.66 ± 0.57 ^b^
27	CAF2	*T. rotundus*		Negative	0 ± 0 ^a^
28	CAF3	*P. purpurogenum*		Positive	10.66 ± 1.52 ^cd^
29	CAF4	*Cladosporium* sp.		Negative	0 ± 0 ^a^
30	CAF5	*Cylindrocephalum* sp.		Positive	7 ± 1 ^b^

Mean values followed by same letters (a, b, c, etc.) are not significantly different according to DMRT at *p* ≤ 0.05.

**Table 2 biomolecules-10-01412-t002:** Estimation of siderophore content produced by endophytic fungi of *Cymbidium aloifolium* using Chrome Azurol S (CAS) liquid assay.

Sl. No	Fungal Isolate	Endophytic Fungi	% Siderophores ± SD; *n* = 3
1	CAR1	*Aspergillus japonicus*	27.02 ± 0.45 ^j^
2	CAR4	*Colletotrichum gloeosporioides*	40.54 ± 0.45 ^l^
3	CAR5	*Trichoderma* sp.	18.91 ± 0.45 ^e^
4	CAR6	*Xylaria* sp.	40.99 ± 0.68 ^l^
5	CAR9	*Penicillium* *citrinum*	25.52 ± 0.68 ^i^
6	CAR11	*Curvularia* sp.	23.87 ± 0.45 ^h^
7	CAR12	*Aspergillus sydowii*	48.94 ± 0.68 ^n^
8	CAR13	*Cladosporium* sp.	35.73 ± 0.68 ^k^
9	CAR14	*Aspergillus terreus*	46.39 ± 0.45 ^m^
10	CAR15	*Alternaria* *alternata*	18.01 ± 0.45 ^de^
11	CAL1	*Penicillium chrysogenum*	56.6 ± 0.68 ^o^
12	CAL2	*Aspergillus sydowii*	13.06 ± 0.45 ^b^
13	CAL4	*Rhizoctonia* sp.	10.35 ± 0.45 ^a^
14	CAL7	*Colletotrichum* *truncatum*	25.22 ± 0.45 ^i^
15	CAL8	*Alternaria alternata*	21.77 ± 0.68 ^g^
16	CAF1	*Fusarium* *oxysporum*	16.06 ± 0.68 ^c^
17	CAF3	*Penicillium purpurogenum*	17.71 ± 0.68 ^d^
18	CAF5	*Cylindrocephalum* sp.	20.41 ± 0.69 ^f^

Mean values followed by same letters (a, b, c, etc.) are not significantly different according to DMRT at *p* ≤ 0.05.

**Table 3 biomolecules-10-01412-t003:** Percentage of Fe^3+^ scavenged by endophytic fungal siderophores.

Sl. No	Fungal Siderophore	Iron as Fe^3+^ by AAS (%)	Percentage of Fe^3+^ with Reference to Control (x)	Percentage of Fe^3+^ Scavenged by Siderophore: (100 − x)
1	Control (2% FeCl_3_)	0.45	100%	-
2	*A. sydowii* (CAR12)	0.13	28.88%	71.12%
3	*P. chrysogenum* (CAL1)	0.076	16.88%	83.12%
4	*A. terreus* (CAR14)	0.12	26.66%	73.34%

**Table 4 biomolecules-10-01412-t004:** Antimicrobial activity of fungal siderophores on *Ralstonia solanacearum* isolated from infected groundnut plants.

Sl. No	*R. solanacearum* Isolate	Zone of Inhibition in mm ± SD; *n* = 3			
		Positive Control	CAR12	CAR14	CAL1
1	APM39	9 ± 1 ^abc^	6.66 ± 0.57 ^b^	6.33 ± 0.57 ^b^	8.66 ± 0.57 ^abc^
2	APM42	11.33 ± 1.15 ^ef^	8.66 ± 0.57 ^cd^	7.66 ± 0.57 ^bcd^	9.66 ± 0.57 ^abcde^
3	APM52	19.66 ± 1.52 ^jk^	9.66 ± 1.52 ^de^	12.33 ± 1.52 ^f^	16.33 ± 1.41 ^hijk^
4	APM53	10.33 ± 1.52 ^bcde^	8.33 ± 0.57 ^cd^	6.66 ± 1.15 ^b^	8 ± 1 ^ab^
5	KAP1	10.66 ± 0.57 ^cde^	0 ^a^	7.33 ± 0.57 ^bc^	8.66 ± 0.57 ^abc^
6	KAP4	9.33 ± 0.57 ^abcd^	0 ^a^	0 ± 0 ^a^	7.66 ± 0.57 ^a^
7	KAP6	8.66 ± 0.57 ^ab^	6.33 ± 0.57 ^b^	7.33 ± 0.57 ^bc^	8 ± 1 ^ab^
8	KAP8	12.66 ± 0.57 ^fg^	8.33 ± 0.57 ^cd^	7.66 ± 0.57 ^bcd^	9.33 ± 0.57 ^abcd^
9	KAP17	8.33 ± 0.57 ^a^	6.66 ± 1.15 ^b^	7.66 ± 1.15 ^bcd^	9 ± 1 ^abcd^
10	KAP18	20 ± 1 ^jk^	10.33 ± 0.57 ^ef^	11.66 ± 0.57 ^f^	17 ± 1 ^ijk^
11	KAP19	10.66 ± 0.57 ^cde^	9.66 ± 0.57 ^de^	8.33 ± 0.57 ^cde^	10.66 ± 0.57 ^abcdef^
12	APH25	18.33 ± 0.57 ^ij^	11.66 ± 0.57 ^fg^	9.66 ± 0.57 ^e^	14.66 ± 0.57 ^ghij^
13	APH26	20.33 ± 0.57 ^k^	14.33 ± 0.57 ^h^	14.66 ± 0.57 ^g^	17.33 ± 1.15 ^jk^
14	APH28	12.66 ± 0.57 ^fg^	9 ± 1 ^cde^	8.66 ± 0.57 ^cde^	11.66 ± 0.57 ^bcdefg^
15	APH36	13.66 ± 0.57 ^gh^	11.33 ± 0.57 ^fg^	8.66 ± 0.57 ^cde^	12.66 ± 0.57 ^defgh^
16	APK9	17.66 ± 0.57 ^i^	14 ± 1 ^h^	14.33 ± 0.57 ^g^	16.33 ± 0.57 ^hijk^
17	APK10	13.66 ± 1.15 ^gh^	11.33 ± 0.57 ^fg^	12.33 ± 0.57 ^f^	13.66 ± 0.57 ^fghij^
18	APA37	18.33 ± 0.57 ^ij^	12.33 ± 0.57 ^g^	14.66 ± 0.57 ^g^	16.66 ± 0.57 ^ijk^
19	APA63	11 ± 1 ^def^	0 ^a^	8.33 ± 0.57 ^cde^	8.33 ± 1.15 ^ab^
20	APP66	22.66 ± 1.52 ^l^	14.33 ± 1.52 ^h^	12.33 ± 0.57 ^f^	19.33 ± 0.57 ^k^
21	APP69	13.66 ± 1.15 ^gh^	9.66 ± 0.57 ^de^	11 ± 0 ^f^	13.33 ± 1.52 ^efghi^
22	APP70	12.66 ± 1.52 ^fg^	0 ^a^	0 ^a^	8.66 ± 1.15 ^abc^
23	APP71	15.33 ± 0.57 ^h^	9.33 ± 0.57 ^de^	9 ± 1 ^de^	11.33 ± 0.57 ^abcdefg^
24	APP73	15.33 ± 1.52 ^h^	7.66 ± 0.57 ^bc^	9 ± 1 ^de^	12.33 ± 0.57 ^cdefg^
25	APP74	12.66 ± 0.57 ^fg^	8.33 ± 1.15 ^cd^	8.66 ± 1.15 ^cde^	12.33 ± 0.57 ^cdefg^

Mean values followed by same letters (a, b, c, etc.) are not significantly different according to DMRT at *p* ≤ 0.05.

**Table 5 biomolecules-10-01412-t005:** Antimicrobial activity of fungal siderophores on *Xanthomonas oryzae* pv*. oryzae* (Xoo) from infected rice plants.

Sl. No	Xoo Isolate	Zone of Inhibition in mm ± SD; *n* = 3			
		Positive Control	CAR12	CAR14	CAL1
1	MBBT01	21.33±1.52 ^cde^	13.66 ± 1.52 ^cdef^	16.33 ± 1.52 ^de^	18.33 ± 0.57 ^ef^
2	MBBT02	23.66 ± 1.52 ^fg^	18.66 ± 1.52 ^k^	18.66 ± 1.15 ^fgh^	21.66 ± 1.52 ^hi^
3	MBBT03	16.33 ± 1.52 ^ab^	12.33 ± 0.57 ^c^	13.33 ± 1.15 ^bc^	14.66 ± 0.57 ^abc^
4	MBBT04	23 ± 1 ^efg^	15.33 ± 0.57 ^fgh^	17.33 ± 0.57 ^ef^	20 ± 1 ^fgh^
5	MBBT05	18 ± 1 ^b^	12.33 ± 0.57 ^c^	13.33 ± 0.57 ^bc^	16.33 ± 0.57 ^cd^
6	MBBT06	16.33 ± 1.52 ^ab^	9.33 ± 0.57 ^b^	11.33 ± 0.57 ^a^	14.33 ± −0.57 ^ab^
7	MBBT07	20.66 ± 0.57 ^cd^	13.66 ± 1.15 ^cdef^	14.33 ± 0.57 ^c^	17.66 ± 1.15 ^de^
8	MBBT08	24 ± 1 ^g^	18.33 ± 1.52 ^jk^	19.33 ± 0.57 ^gh^	21.66 ± 1.52 ^hi^
9	MBBT09	18 ± 1 ^b^	12.66 ± 0.57 ^cd^	13.66 ± 0.57 ^bc^	15.33 ± 0.57 ^bc^
10	MBBT10	23.66 ± 1.52 ^fg^	16.66 ± 1.15 ^hij^	20.33 ± 0.57 ^h^	22 ± 1 ^i^
11	MBBT11	16.33 ± 1.52 ^ab^	9 ± 1 ^ab^	12.33 ± 0.57 ^ab^	14.66 ± 0.57 ^abc^
12	MBBT12	14.66 ± 0.57 ^a^	7.33 ± 0.57 ^a^	12.33 ± 0.57 ^ab^	13.33 ± 0.57 ^a^
13	MBBT13	17.33 ± 0.57 ^b^	8.66 ± 0.57 ^ab^	11.33 ± 1.52 ^a^	15.33 ± 0.57 ^bc^
14	MBBT14	23.66 ± 1.15 ^fg^	17 ± 1 ^hijk^	18 ± 1.73 ^efg^	19.66 ± 0.57 ^fg^
15	MBBT15	24.33 ± 0.57 ^g^	17.66 ± 2.08 ^jk^	18.66 ± 1.15 ^fgh^	23.66 ± 1.15 ^j^
16	MBBT16	22.66 ± 0.57 ^defg^	16.33 ± 1.52 ^ghi^	17.66 ± 0.57 ^efg^	18.66 ± 0.57 ^ef^
17	MBBT17	22.66 ± 1.15 ^defg^	12.33 ± 0.57 ^c^	16.33 ± 1.52 ^de^	20 ± 1 ^fgh^
18	MBBT18	22.66 ± 0.57 ^defg^	9.66 ± 0.57 ^b^	14.66 ± 1.52 ^cd^	18.33 ± 0.57 ^ef^
19	MBBT19	20.66 ± 2.08 ^cd^	14.66 ± 0.57 ^efg^	16.33 ± 0.57 ^de^	17.66 ± 1.52 ^de^
20	MBBT20	20.33 ± 1.52 ^c^	13.33 ± 0.57 ^cde^	14.66 ± 0.57 ^cd^	17.33 ± 1.15 ^de^
21	MBBT21	23.66 ± 0.57 ^fg^	15.33 ± 0.57 ^fg^	17.33 ± 057 ^ef^	20.66 ± 0.57 ^ghi^
22	MBBT22	21.66 ± 0.57 ^cdef^	12.33 ± 0.57 ^c^	13.66 ± 0.57 ^bc^	17.33 ± 1.15 ^de^
23	MBBT23	14.66 ± 0.57 ^a^	7.33 ± 0.57^a^	12.33 ± 0.57^ab^	13.33 ± 0.57^a^
24	MBBT24	21.33 ± 1.52 ^cde^	14.33 ± 1.52 ^ef^	16.33 ± 1.52 ^de^	18.66 ± 0.57 ^ef^
25	MBBT25	23 ± 1 ^efg^	15.33 ± 0.57 ^fgh^	17.33 ± 0.57 ^ef^	20 ± 1 ^fgh^

Mean values followed by same letters (a, b, c, etc.) are not significantly different according to DMRT at *p* ≤ 0.05.

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
