# Peer review of "Detection and Characterization of Antibacterial Siderophores Secreted by Endophytic Fungi from Cymbidium aloifolium"

_biomolecules, 2020, doi:10.3390/biom10101412_

Round 1
Reviewer 1 Report
The manuscript from Chowdappa describes the production of siderophores in endophytic fungi of Cymbidium aloifolium. The optimisation of siderophores production by endophytes of Cymbidium aloifolium are illustrated as well the identification of siderophores with antimicrobial activity. The work would represent a significant piece of work to the scientific community, particularly in the area of endophytes and plants and microbiome communication.
I suggest a brief description of the methods. It is OK to mention them in references, but it will help to the reader to follow and understand easily the logic of the methods proposed by the authors.
The introduction requires a little bit more explanation of the nature of siderophores.
Describe the full name of the methods from lines 70 and 71.
Homogenesie the quality of the panels from Figure 4. In some cases the quality of the panels is low, or the axes titles look different in each panel, in some cases stretched and in some other compacted.
The reference 36 (Storey, M.Sc. thesis) needs to be substituted by the published manuscript from the same author: Storey, E. P., et al. "Characterisation of ‘Schizokinen’; a dihydroxamate-type siderophore produced by Rhizobium leguminosarum IARI 917." Biometals 19.6 (2006): 637-649.
Author Response
Responses to the Reviewers’ Comments
The authors would like to thank the Reviewer’s for his/her constructive comments and suggestions that have helped us improve our manuscript. An extensive revision has been undertaken and incorporated all the corrections and suggestions raised by the Reviewer’s in the revised manuscript.
Reviewer 1:
The manuscript from Chowdappa describes the production of siderophores in endophytic fungi of Cymbidium aloifolium. The optimisation of siderophores production by endophytes of Cymbidium aloifolium are illustrated as well the identification of siderophores with antimicrobial activity. The work would represent a significant piece of work to the scientific community, particularly in the area of endophytes and plants and microbiome communication.
Response: We would like to express our special thanks to the Reviewer for positively evaluated our manuscript. All the comments were addressed objectively and corrections were incorporated in the revised manuscript.
Comment 1: I suggest a brief description of the methods. It is OK to mention them in references, but it will help to the reader to follow and understand easily the logic of the methods proposed by the authors
Response: Thank you very much for this suggestion with which we totally agree. As per the Reviewer suggestion, a brief description of the methodology along with the appropriate citation/s has been incorporated in the revised manuscript (L.92-100, 114-115, 124-136 and 183-189).
Comment 2: The introduction requires a little bit more explanation of the nature of siderophores.
Response: We highly appreciate the Reviewer for his/her critical observation of our manuscript. As suggested by the Reviewer, we have now included the below paragraph explaining the nature of siderophores in the Introduction section of the revised manuscript.
Comment 3: Describe the full name of the methods from lines 70 and 71.
Response: Thank you very much for this suggestion. The sentence is now completely revised for easy understanding of the key objectives of the study (L.76-84).
Comment 4: Homogenesie the quality of the panels from Figure 4. In some cases the quality of the panels is low, or the axes titles look different in each panel, in some cases stretched and in some other compacted.
Response: We thank the Reviewer for this critical observation. The Figure 4 is now replaced as Figure 2 with a high resolution (600 dpi) and all the errors are fixed in revised Figure 2 (P.9).
Comment 5: The reference 36 (Storey, M.Sc. thesis) needs to be substituted by the published manuscript from the same author: Storey, E. P., et al. "Characterisation of ‘Schizokinen’; a dihydroxamate-type siderophore produced by Rhizobium leguminosarum IARI 917." Biometals 19.6 (2006): 637-649.
Response: Thank you very much for this comment and suggestion. To meet the Reviewer suggestion, the above suggested citation has been replaced in the revised manuscript (L.611-613).
Reviewer 2 Report
This is a good piece of work looking at endophytes of an orchid their use as antimicrobial agents. However, there are some major points to streamline to include the data in the manuscript. To start with the title itself to be changed- “Detection……………………..from endophytic Cymbidium aloifolium.” It must be “…….. endophytic fungi from Cymbidium aloifolium”
The first sentence in abstract is not correct. Dis they mean “…………………………. large number of secondary………………….”
Rather than saying ectendo (line no 51) they must say endophytic as that is the group they are discussing here. I would like the authors to look at their main finding and rewrite the paper and discuss only the major points.
The objectives in the introduction need to be clear and that must be the points they have to include when they summarise their findings. There is some disconnect noted here. Quite a lot of peripheral analyses can be included in the supplemental data. Although this may be useful it may not add more value to the paper. This means it will cut out lengthy discussion included in the manuscript.
Identification of endophytes from any plant, especially from orchids has routine procedures. This looks like a culture-dependant method. But the identification is done not based on ITS sequencing or similar methods. This throws a lot of doubt into what kind of fungus it is. As most of this dependant on specific genotypes the fungi used for this study required a bit more clarity on their taxonomic status. I suggest the authors to do that work.
As I understand there are changes based on different culture conditions on siderophore production. Interactions must be analysed using an appropriate statistical method. Once these points are addressed the paper can be revised and resubmitted.
Author Response
Responses to the Reviewers’ Comments
The authors would like to thank the Reviewer’s for his/her constructive comments and suggestions that have helped us improve our manuscript. An extensive revision has been undertaken and incorporated all the corrections and suggestions raised by the Reviewer’s in the revised manuscript.
Reviewer 2:
This is a good piece of work looking at endophytes of an orchid their use as antimicrobial agents. However, there are some major points to streamline to include the data in the manuscript. To start with the title itself to be changed- “Detection……………………..from endophytic Cymbidium aloifolium.” It must be “…….. endophytic fungi from Cymbidium aloifolium
Response: We are very glad that the Reviewer highly evaluated our manuscript, and provided constructive comments and suggestions for us to improve our manuscript.
As recommended by the Reviewer, the title is revised as follows:
“Detection and Characterization of Antibacterial Siderophore Secreted by Endophytic fungi from Cymbidium aloifolium” (L.2-4).
Comment 1: The first sentence in abstract is not correct. Dis they mean “…………………………. large number of secondary…..
Response: We thank the Reviewer for this critical observation. We have now revised the first sentence of the abstract as follows:
“Endophytic fungi from orchid plants are reported to secrete secondary metabolites which involve bioactive antimicrobial siderophores. Hence, in this study endophytic fungi capable of secreting siderophores were isolated from a medicinal orchid plant, Cymbidium aloifolium” (L.18-20).
Comment 2: Rather than saying ectendo (line no 51) they must say endophytic as that is the group they are discussing here. I would like the authors to look at their main finding and rewrite the paper and discuss only the major points
Response: We highly appreciate the Reviewer for this comment and suggestion. To meet the Reviewer suggestion, the sentence is revised as follows:
“Majority of the siderophores are reported to be endophytic in nature found in various microbial origin including mycorrhiza and orchidaceous fungi [8, 10, 11]” (L.52-53).
In addition, as the Reviewer rightly pointed out we have now briefly explain our key findings of the work with the other published work in the discussion section of revised manuscript.
Comment 3: The objectives in the introduction need to be clear and that must be the points they have to include when they summarise their findings. There is some disconnect noted here. Quite a lot of peripheral analyses can be included in the supplemental data. Although this may be useful it may not add more value to the paper. This means it will cut out lengthy discussion included in the manuscript
Response: The authors would like to thank the Reviewer for this important remark/s. As recommended by the Reviewer, we have now revised the last section of the introduction which is focused on the main objective of this work as below:
“In recent years, research is mainly focused on identification and characterization of bioactive siderophores from endophytic fungi. To our knowledge this is the first report on siderophore production from the endophytic fungi isolated from a medicinal orchid plant, Cymbidium aloifolium. Hence, the present investigation is focused on (i) isolation of endophytic fungi associated with different parts of C. aloifolium, (ii) optimization of culture conditions for the production of siderophores from endohytic fungi, (iii) detection and quantification of siderophore production from isolated endophytic fungi, (iv) identification and purification of type of siderophores and (v) bioactivity of siderophores against Ralstonia solanacearum and Xanthomonas oryzae pv. Oryzae” (L.76-84).
Also, to avoid confusion, few data figures were shifted to supplementary figures. We now believe the discussion of the revised manuscript is reduced and reaches the audience more suitably.
Comment 4: Identification of endophytes from any plant, especially from orchids has routine procedures. This looks like a culture-dependant method. But the identification is done not based on ITS sequencing or similar methods. This throws a lot of doubt into what kind of fungus it is. As most of this dependant on specific genotypes the fungi used for this study required a bit more clarity on their taxonomic status. I suggest the authors to do that work
Response: Thank you so much again for raising this comment. Yes, we agree with your views. The isolation of the endophytes was carried out using standard procedure as follows:
“Various plant parts (leaf, root and flower) were harvested from C. aloifolium and surface sterilized using sodium hypochlorite (0.2%), following by three washes with sterile distilled water. The dried plant parts were used for isolation of endophytic fungi on Potato Dextrose Agar (PDA) plate amended with tetracycline (50 µg/mL) and sterilized plant parts incubated on PDA without tetracycline as appropriate control Zhu et al. [24] (L.92-96).
This follows the identification of endophytic fungi capable of producing siderophores using ITS region sequencing (ITS1 and ITS4 primers). The obtained sequences were assembled and blast search for their similarity or homology by NCBI. The authenticated names of the fungal species were submitted to GenBank with accession numbers - KX553900, KX553901 and KX553902 respectively (L.228-231).
To determine the taxonomic status and relation with the existing fungi we have now draw the phylogentic dendrogram tree and incorporated as Supplementary Figure (L.231-233).
Comment 5: As I understand there are changes based on different culture conditions on siderophore production. Interactions must be analysed using an appropriate statistical method. Once these points are addressed the paper can be revised and resubmitted
Response: We appreciate the Reviewer for this query with which we totally agree. We tried to correlate the interactions siderophore production with various culture parameters using pearson correlation plot. However, the cultural parameters have different variables and could not process the data. Hence, the correlation analysis of the siderophore production based on various culture conditions were statistically performed with ANOVA and Duncan Multiple Range Test (DMRT) test using SPSS software (version 20). The significance differences were measured based on P value (< 0.05). The obtained data are represented as Figure 2 in the revised manuscript (P.9).

Round 2
Reviewer 2 Report
The letter the authors submitted is not a permit from the relevant authorities for collecting rare plant material from Western Ghats. This is unfortunate authors submitted a note confirming the identification of the sample they collected. It is important for you to know the regulations put in place by the relevant bodies and follow those. They are in place for a reason and this has not been followed in this case.
This is not the first report of siderophores from fungi in orchids. Medicinal orchid is an arbitrary term, therefore it will be better to remove this claim. The authors have already included the following reference in the manuscript.
Haselwandter, K. 2008 Structure and function of siderophores produced by mycorrhizal fungi. Mineralogical Magazine72(1), pp. 61-64
Author Response
Comment 1: The letter the authors submitted is not a permit from the relevant authorities for collecting rare plant material from Western Ghats. This is unfortunate authors submitted a note confirming the identification of the sample they collected. It is important for you to know the regulations put in place by the relevant bodies and follow those. They are in place for a reason and this has not been followed in this case.
Response: We would like to express our special thanks to the Reviewer for this query. In India the National Research Centre for Orchids is situated at Sikkim and also has several regional stations including few recognized centres. Among the recognized centres, orchidarium house located at Department of Botany, Bangalore University, India is an orchium collection and maintenance centre headed by Prof. M.C. Gayathri (Certified copy is enclosed below with the images of orchidarium house). Orchids collected from various regions of southern India are usually maintained in this centre from 2002. In addition, the orchid plant (Cymbidium aloifolium) used in this study is used as traditional herbal medicine by the local peoples of southern India. In Karnataka, the Regional Ayurveda Research Institute for Metabolic disorders, Central Council for Research in Ayurvedic Sciences, Ministry of AYUSH, Government of India is the authorized authority for the all kinds of medicinal plants for taxonomix identification. http://www.ccras.nic.in/content/regional-ayurveda-research-institute-metabolic-disorders-bangalore.
The orchidarium head Prof. M.C. Gayathri had issued an official letter corresponding to the Editor of the esteemed Journal “Biomolecules” that is enclosed below:
Orchidarium house, Bangalore University
.
Comment 2: This is not the first report of siderophores from fungi in orchids. Medicinal orchid is an arbitrary term, therefore it will be better to remove this claim. The authors have already included the following reference in the manuscript
Haselwandter, K. 2008 Structure and function of siderophores produced by mycorrhizal fungi. Mineralogical Magazine72 (1), pp 61-64.
Response: Thank you so much for this comment with which we totally agree. To meet the Reviewer suggestion, the sentence claiming first report of siderophores from endophytic fungi isolated from orchis plant from the introduction section is deleted from the revised manuscript.
Comment 3:Introduction and methods – Needs improvement
Response: We appreciate this Reviewer for critical observation of our manuscript. An extensive revision has been undertaken in the Introduction and Methods section for easy understanding of the contents to the audience.
